# Geostatistical Modeling and Heterogeneity Analysis of Tumor Molecular Landscape

**DOI:** 10.3390/cancers14215235

**Published:** 2022-10-25

**Authors:** Morteza Hajihosseini, Payam Amini, Dan Voicu, Irina Dinu, Saumyadipta Pyne

**Affiliations:** 1School of Public Health, University of Alberta, Edmonton, AB T6G 1C9, Canada; 2Department of Biostatistics, School of Public Health, Iran University of Medical Sciences, Tehran 14155-6559, Iran; 3Faculty of Engineering, McGill University, Montreal, QC H3A 0G4, Canada; 4Health Analytics Network, Pittsburgh, PA 15237, USA; 5Department of Statistics and Applied Probability, University of California, Santa Barbara, CA 93106, USA

**Keywords:** spatial single-cell analysis, intratumor heterogeneity, kriging, spatial entropy, Wasserstein distance, cancer, RNA-seq

## Abstract

**Simple Summary:**

The present study introduces a new computational platform referred to as GATHER to conduct Geostatistical Analysis of Tumor Heterogeneity and Entropy in R. GATHER has several distinct advantages such as (a) a novel use of single-cell-specific spatial information for kriging to synthesize high-resolution and continuous gene expression landscapes of a given tumor sample, (b) the integration of such landscapes to identify and map the enriched regions of the pathways of interest, (c) the identification of genes that have a spatial differential expression at locations representing specific phenotypic contexts, (d) the computation of spatial entropy measures for quantification and objective characterization of intratumor heterogeneity, and (e) the use of new tools for the insightful visualization of spatial transcriptomic phenomena.

**Abstract:**

Intratumor heterogeneity (ITH) is associated with therapeutic resistance and poor prognosis in cancer patients, and attributed to genetic, epigenetic, and microenvironmental factors. We developed a new computational platform, GATHER, for geostatistical modeling of single cell RNA-seq data to synthesize high-resolution and continuous gene expression landscapes of a given tumor sample. Such landscapes allow GATHER to map the enriched regions of pathways of interest in the tumor space and identify genes that have spatial differential expressions at locations representing specific phenotypic contexts using measures based on optimal transport. GATHER provides new applications of spatial entropy measures for quantification and objective characterization of ITH. It includes new tools for insightful visualization of spatial transcriptomic phenomena. We illustrate the capabilities of GATHER using real data from breast cancer tumor to study hallmarks of cancer in the phenotypic contexts defined by cancer associated fibroblasts.

## 1. Introduction

In their well-known paper in 2010, Hanahan and Weinberg noted that tumors exhibit an additional dimension of complexity through their “tumor microenvironment” that contributes to the acquisition of the so-called hallmark traits of cancer. Indeed, extensive studies in recent decades have uncovered a great diversity of cell populations in tumors, thus leading to the active research area of intratumor heterogeneity (ITH) [1]. It has been found to be associated with poor prognosis, therapeutic resistance, and treatment failure leading to poor overall survival in cancer patients [2,3,4,5,6]. ITH is attributed to genetic, epigenetic, and microenvironmental factors [1,7]. Tumors can develop a resistance to treatment due to ITH by new genetic mutations, recovering functionality of previously inactivated genes, phenotypic changes, and variations in response to the microenvironment [8,9].

The persistence of some of the drug-tolerant intratumor cell populations could be attributed to their high phenotypic plasticity. While hierarchies of differentiation also exist among normal cells in healthy tissues, the populations of tumor cells, in contrast, display a far greater cell-to-cell variability and the resulting phenotypic instability [10,11]. Such ITH could be attributed to genetic causes ranging from aneuploidy to spontaneous cell fusions, for example, between cancer and non-cancer cells, in addition to other factors such as complex contextual signals in the highly aberrant tumor microenvironments, or even global alterations in cancer cell epigenomes [12]. ITH also involves immune cell infiltration, which is of evident importance for immunotherapies. Tumor antigen diversity could be determined by the T cell clonality in the different regions of the same tumor [13]. Studies have shown spatially complex interactions between tumor microenvironments and the patient’s immune system [14,15].

While heterogeneous cell types are prevalent within the tumor microenvironment, and while some of them may account for cancer development and progression, it also contains different non-malignant components, including the cancer-associated fibroblasts (CAFs) [16,17,18]. Although the origin and activation mechanism of CAFs remains an area of active research [19,20,21,22], studies have attributed the processes of formation and derivation of CAFs to various precursor cells [20,23], which may be the source of the well-known heterogeneity among the CAFs [24,25,26,27]. Indeed, certain tumors, such as in the breast, in which the prevalence of CAFs can be as high as 80%, they can play both anti- as well as pro-tumorigenic roles [28,29,30]. Importantly, CAFs can facilitate drug resistance dynamically by altering the cell–matrix interactions that control the outer layer of the cells’ sensitivity to apoptosis, producing proteins that control cell survival and proliferation, assisting with cell–cell communications, and activating epigenetic plasticity in neighboring cells [31,32].

In order to understand the spatial heterogeneity of gene expressions, including drug target genes, different sites of the same tumor were analyzed with multiregional RNA sequencing for different cancers [5,33,34,35]. It was observed, for instance, that if HER2+ breast tumors expressed HER2 uniformly across their cells, then the known HER2-targeted therapies were highly effective; and if not, then such patients were found to have shorter disease-free survival [36]. In recent years, with a higher resolution, tissue-specific gene expression analysis is made possible by using new platforms such as single-cell RNA sequencing (scRNA-seq), which has rapidly evolved as a powerful and popular tool [37,38]. This has led to several scRNA-seq studies of the composition of CAFs in different stages of cancer [39,40,41,42,43,44,45,46,47]. For instance, the Human Tumor Atlas Network (https://humantumoratlas.org (accessed on 20 September 2022)) is increasingly enriched with data on human cancers based on scRNA-seq assays.

Despite the advancements and efficacy of scRNA-seq, the lack of spatial information in a scRNA-seq analysis is a significant shortcoming for typical scRNA-seq methods to capture cellular heterogeneity. On the other hand, while oncologic pathologists have long studied cell signaling within tumors by the manual scoring of discordance between individuals and variation between different batches using tissue immunostaining and microscopy, such techniques typically allow for only a few selected markers to be observed per cell, and thus offer a limited reporting of the extent of the potential heterogeneity. Combining high-resolution gene expression data with spatial coordinates can resolve these experimental shortcomings [48]. For instance, spatial proximity to fibroblasts has been shown to impact molecular features and therapeutic sensitivity of breast cancer cells, influencing clinical outcomes [49]. While imaging the transcriptome in situ with high accuracy has been a major challenge in single-cell biology, the development of high-throughput platforms for sequential fluorescence and in situ hybridization such as RNA seqFISH+ and algorithms such as CELESTA have made it possible to identify cell populations and their spatial organization in intact tissues [5,50].

In order to approach and conceptualize the diversity in the spatial omics information, the concept of a habitat and its locations have been studied in association with genetic heterogeneity in a tumor [51,52]. In fact, it was noted that the spatial distribution of genetically distinct tumor cell populations may be correlated with poor clinical outcomes [9]. Landscape ecology is, therefore, a potentially effective modeling framework which—similar to the modeling of an ecosystem’s behavior in terms of the actions and interactions of the individuals and groups of the different constituent species—could be adopted to study the spatio-temporally dynamic and heterogeneous system that is often represented by a tumor.

The present study introduces a new computational platform referred to as GATHER to conduct Geostatistical Analysis of Tumor Heterogeneity and Entropy in R. GATHER uses geostatistical modeling and spatial entropy measures for the quantification and objective characterization of intratumor heterogeneity, as well as to identify different transcriptomic patterns in the molecular landscapes of a tissue sample. Geostatistical models provide a well-established theoretical framework for the prediction and interpolation of spatial data. Kriging, for example, is a generalized least-square regression approach to predict spatial attributes at unobserved locations [53]. GATHER applies kriging for estimating gene-specific, and thereby gene set-specific, expression values at every point of the given tumor space. By constructing such continuous molecular landscapes, it allows for the visualization and identification of local and regional transcriptomic variations. Furthermore, GATHER provides a quantitative characterization of ITH based on spatial entropy measures [54]. Finally, GATHER applies a Wasserstein distance based on a 2-sample test, which is adapted specifically for use on single-cell data [55], to provide two different approaches to identify genes that have spatial differential expressions either (i) near a specific location in the tumor space versus elsewhere, or (ii) across different regions in which a selected phenotypic context is present at different levels.

The concept of entropy in information theory, defined as strings of symbols by Claude Shannon in 1948 [56], has been adapted and used in various contexts due to its ability to capture a broad set of notions such as information content, unexpectedness, uncertainty, diversity, and contagion [57]. Indeed, it was shown that the cancer epigenome has a higher entropy than its normal counterpart [10]. In the present study, we are more specifically interested in the spatial entropy of a tumor’s molecular information content. Despite the early applications of Shannon’s entropy (H) to evaluate spatial heterogeneity in geographical phenomena [58] and landscape ecology [59], researchers have noted that, while H takes into account the number of symbols of each type in a string, it ignores the effect of their spatial arrangement [60]. This has led to the further development of different entropy measures that specifically include spatial information. In particular, the present study computed Batty’s entropy to measure the spatial heterogeneity of diverse phenotypes in the tumor space, as well as Leibovici’s co-occurrence-based entropy for heterogeneity of a given gene set’s enrichment in a particular phenotypic context.

In 2020, a paper listed eleven grand challenges in single-cell data science, which included the challenge of “finding patterns in spatially resolved measurements” [61]. Towards this, many recent efforts have produced computational methods to analyze spatial information in single-cell studies [62,63,64,65,66,67,68,69,70]. The aim of the present study is to address the said challenge using a different (geostatistical modeling) approach in comparison to the existing ones. This gives GATHER several distinct advantages such as (a) the use of single-cell-specific spatial information for kriging to synthesize high-resolution and continuous gene expression landscapes of a given tumor sample, (b) the integration of such landscapes to identify and map the enriched regions of the pathways of interest, (c) the identification of genes that have a spatial differential expression at locations representing specific phenotypic contexts, (d) the computation of spatial entropy measures for the quantification and objective characterization of ITH, and (e) the use of new tools for the insightful visualization of spatial transcriptomic phenomena. In the next section, we describe the data and methods, followed by the results of real tumor data analysis using GATHER, and end with a discussion including future work.

## 2. Data and Methods

### 2.1. Data

The spatial transcriptomics data were downloaded from the 10x Genomics online resource (Available at: https://www.10xgenomics.com/resources/datasets/human-breast-cancer-whole-transcriptome-analysis-1-standard-1-2-0 (accessed on 15 December 2020). The data were generated using the Visium Spatial Gene Expression protocol run on an invasive breast cancer tissue sample that is estrogen receptor (ER)-positive, progesterone receptor (PR)-positive, and human epidermal growth factor receptor 2 (HER2)-negative. RNA sequencing data were generated with a paired-end, dual-indexed process using Illumina NovaSeq 6000, with a sequencing depth of 72,436 mean reads per cell. After filtering the downloaded dataset for average gene expression value >1, the data matrix contained 1981 rows (genes) and 4325 columns (single cells). Figure 1 shows the steps of the GATHER workflow which includes spatial gene expression analysis. As part of our RNAseq data preprocessing, the zero counts were replaced with a small random jitter around zero that would minimally affect the remaining gene expression values. We normalized each gene’s expression across samples with a 10-fold cross-validation-based data transformation method using bestNormalize package in R software [71].

### 2.2. Constructing Gene Expression Landscape by Kriging

Our dataset is defined on a 2-dimensional tissue space, with a specified coordinate system. We discretized this space using an evenly spaced grid of size 80×80, i.e., 6400 unique point locations over a rectangular area covering 50 units below (above) the minimum (maximum) values of x and y coordinates of the cells in our dataset.

In this study, the geostatistical method of ordinary kriging (OK) was used to interpolate the expression value of each gene g at each grid point p based on the best linear unbiased prediction (blup) using a weighted average expression of g in the cells that lie in a given neighborhood of p. The basic model for the OK predictor (Waller and Gotway, 2004) of the expression $Z\left(g,\;s_{0}\right)$ of g at an arbitrary location $s_{0}$ in the given tissue space is computed as:$$\hat{Z}\left(g,\;s_{0}\right)=\sum_{i=1}^{N_{g}}\lambda_{g,i}Z\left(g,\;s_{i}\right)$$
where $Z\left(g,s_{i}\right)$ is the normalized expression value of g in cell i at the location $s_{i}$, $\lambda_{g,i}$ is the weight attributed to the measured expression of g at location $s_{i}$, and $N_{g}$ is the number of available single-cell measurements of the expression of g. For OK, we assume stationary Z(g, ⋅) and a known semivariogram (of g). The kriging weights that determine the contributions of the measurements are defined by an empirical semivariogram function that describes the spatial dependence among the single-cell expression values of g in terms of inter-cellular distance [53]. Typically, such contribution to the kriged expression value at $s_{0}$ decrease for a cell $s_{i}$ as it moves farther from $s_{0}$. GATHER also computes the kriging standard error [72] at the same location $s_{0}$, which gives a measure of the uncertainty of the prediction of $Z\left(g,s_{0}\right)$. Thus, GATHER constructs gene-specific, continuous transcriptomic landscapes, along with the maps of the corresponding standard errors, which could be visualized for each gene separately (or as spatially combined for a given gene set), an example of which is shown in Figure 1.

### 2.3. Test of Spatial Differential Expression of Genes

Our platform allows us to identify a spatial phenotype in terms of differential expression of one or more “marker” genes that are known to characterize the phenotype. This allows us to demarcate and map the regions in the tissue space where the phenotype is significant. To map the co-occurrence of more than one phenotype, distinct colors were used. Furthermore, the presence of these spatial phenotypes could serve as specific contexts within which certain genes of interest may show a differential expression. Indeed, we developed methods for identifying such genes as well as measuring the contextual enrichment of gene sets and curated molecular pathways.

Differentially expressed genes were detected using the semi-parametric 2-Wasserstein distance test for single-cell data [55]. In this study, the test was applied in a spatially contextualized manner using two different approaches. In our first approach, we identified and mapped the significance of local expression of a given gene at any point of the tissue space, which is systematically discretized by a well-defined grid (see above). For this purpose, we begin by grouping the cells that are local to a given location and distinguish them from the group of nonlocal cells that are distant from this location. At each point p of the grid, we defined a neighborhood Nbd(p,r) centered at p and based on a circle of radius r. The value of r is chosen to be the 25th percentile of all pairwise distances between the cells, thus ensuring proximity among the cells that lie in Nbd(p,r). The cells that lie in Nbd(p,r) are called “local”, and the rest are termed “non-local”.

In our second approach, the entire tissue space was partitioned into regions according to different levels of enrichment of a phenotype of interest, e.g., characterized by expression (z) of a marker of cancer-associated fibroblasts (CAFs). The regions of the landscape are thus marked by pre-determined l (=3) discrete levels of the selected phenotype: high (CAFs z>1), mid (CAFs 0.5<z≤1), and low (CAFs z≤0.5). These regions provide the graded spatial contexts in which certain genes may express. Thus, we used the 2-Wasserstein distance method to compare the single-cell level expression of each gene across successive levels of the phenotype, i.e., across (a) the high and the medium regions; and (b) the medium and the low regions. For a selected phenotype, the significantly differentially expressed genes are identified by permutation testing (with 100 repetitions) at a pre-determined FDR adjusted q-value level (for example, 0.2) [73].

### 2.4. Spatial Analysis of Hallmark Gene Sets of Cancer

#### 2.4.1. Gene set Enrichment Landscape Construction

Let L be the list of genes whose expressions are measured (and thus available as spatial z-scores) in the present study.

For a selected gene set S, we computed the spatial enrichment z-score, SEZ(S,p), at each grid point p using the Stouffer’s sum over the kriged value, Z(g, p), of expression of each gene g in S and L at p as follows:$$SEZ\left(S,p\right)=\sum_{g\epsilon S\cap L}Z\left(g,\;p\right)/\surd \left|S\cap L\right|.$$

This allows us to construct a gene set enrichment landscape, which extends the idea of a single-gene expression landscape.

#### 2.4.2. Cancer Hallmark Gene Set Enrichment

We downloaded gene sets (Appendix A) from the Molecular Signatures Database (MSigDB) that represent commonly known “hallmarks” of cancer [74]. To ensure their relevance as well as non-redundancy, we selected 8 of those hallmark gene sets that have at least 25% of overlap with the expressed genes (see above text on preprocessing) but a mutual gene set overlap of less than 10%.

### 2.5. Spatial Entropy of a Tumor Sample

#### 2.5.1. Calculating Phenotypic Diversity

Given our interest to characterize the heterogeneity of a given tumor sample in terms of the spatial phenotypes therein, Batty’s entropy measure was computed to evaluate the distribution of a candidate phenotype over the given tissue area by allowing for its partitioning into subareas of different sizes and shapes. Let a tissue area of size A partitioned into *G* subareas of size $A_{j},\;j=1,\;\ldots G$. If a phenotype of interest F occurs in A, and in $A_{j}$ with probability $p_{j}$, then $\overset{G}{\underset{j=1}{\sum }}p_{j}=1$. The phenotype intensity in $A_{j}\;$ is given by $\lambda_{j}=p_{j}/A_{j}$.

Batty’s spatial entropy for phenotype F occurring over a tissue area A that is randomly partitioned into G subareas is defined as:$$H_{B}\left(F\right)=\sum_{j=1}^{G}p_{j}\log\left(1/\lambda_{j}\right).$$

The maximum value of spatial entropy is log(A)  when F occurs with equal intensity ($\lambda_{j}=1/A$) over all G subareas partitioning the tissue area of size A. The spatial entropy attains a minimum value of $\log\left(A_{j*}\right)$ when the entire F is concentrated in the smallest subarea of size $A_{j*}$. Since the location and size of such subareas are unknown for the occurrence of an arbitrary phenotype, we randomly partitioned the landscape of the tumor to compute Batty’s entropy over different values of G (G=2,3…12), and repeated the partitioning process (N=100 times for each value of G) to output the median $H_{B}\left(F\right)$ as the final measure of spatial heterogeneity of F over A.

#### 2.5.2. Heterogeneity of Gene Set Enrichment in a Phenotypic Context

Batty’s spatial entropy of a variable X can be extended to a co-occurrence-based entropy measure defined using a new categorical variable Z that takes values in the form of ordered pairs $\left(x_{i},x_{j}\right)$ of X which is considered co-occurrent if their distance is less than or equal to a pre-determined threshold d. Given the I categories of X, there are $I^{2}$ categories of Z. As noted by Altieri et al. (2018) [75], an entropy measure based on Z is useful when the variable of interest has multiple categories and the aim is to understand how a spatial context (e.g., a local phenotype’s enrichment in a tumor) may influence its neighborhood outcomes (for example, a selected molecular pathway’s expression). The discretized levels of a given (phenotype, gene set) pair (a realization of Z) at the observed locations could be viewed as multi-categorical point data and their co-occurrence based Leibovici’s spatial entropy (Leibovici et al., 2009) [76] is defined as follows:$$H_{L}\left(Z|d\right)=\sum_{r=1}^{I^{2}}p(z_{r}|d)\log\left(1/p(z_{r}|d)\right).$$

### 2.6. Software

All statistical analyses were performed in R version 4.0.4. We used the Seurat package [77] for data preparation, bestNormalize [78] for normalization, automap [79] for making a standard grid and applying ordinary kriging, waddR [80] to detect differentially expressed genes based on the 2-Wasserstein distance test, and SpatEntropy package [75] for Batty’s and Leibovici’s spatial entropy calculations. The 3-dimensional and interactive plots were generated with plot3D and plotly packages [81,82]. For comparative analysis, we used RNAseq data analysis methods, edgeR [83] and DESeq2 [84]. For the validation of GATHER’s results, a pathway over-representation analysis (based on KEGG pathways) was conducted using the online Database for Annotation, Visualization, and Integrated Discovery (or DAVID—https://david.ncifcrf.gov/ (accessed on 15 September 2022) platform.

## 3. Results

The present study yields a new computational tool, GATHER, for the geostatistical modeling and heterogeneity analysis of molecular landscapes in tumors and tissue samples. The different modules of GATHER are outlined in Figure 1. These include (1) gene-specific expression landscape construction via kriging-based geostatistical prediction, (2) estimating a measure of uncertainty associated with the kriging predictions, (3) computing the spatial gene set enrichment score, (4) identifying genes with spatial differential expression at selected phenotypic contexts, (5) identifying genes with spatial differential expression at selected locations, (6) computing Batty’s spatial entropy to measure phenotypic heterogeneity, and (7) computing Leibovici’s co-occurrence-based entropy to quantify the heterogeneity of a selected gene set’s enrichment in a given phenotypic context. Furthermore, GATHER provides tools for the insightful visualization such as 2D and 3D normalized gene expression landscapes and the corresponding maps of standard errors, spatial enrichment surface of a gene set, as well as spatial entropy-associated diagrams.

We begin with an illustration of the gene-specific expression landscape construction via kriging-based geostatistical prediction. In a past study of 100 breast tumors to understand the complexity of intratumor genetic heterogeneity, driver mutations were observed in several cancer genes [85]. For instance, *TBX3*, which encodes for the transcription factor T-box 3 (TBX3), was found to be overexpressed in different types of carcinomas, including breast cancer. TBX3, a mostly cytoplasmic protein in both normal and breast cancer tissues, is significantly overexpressed in the latter, and thus, could serve as a potential diagnostic marker of breast cancer cells [86]. Yet, TBX3 localizes differently depending on its role and the cell-cycle phase [87]. In order to gain insights into the possible spatial distribution, we used GATHER to construct the expression landscape of *TBX3*, which, along with the corresponding standard errors of the local kriging predictions, are mapped and shown in Figure 2.

Notably, the geostatistical modeling-based transcriptomic landscapes could also be viewed using the 3D interactive visualization tool of GATHER (Figure 2C). Using a grid of evenly spaced points defined on the input tissue space, the z dimension depicts the level of predicted gene expression at each point (x, y) of the synthesized landscape. The interactive 3D visualization tool could be useful for operations such as zooming in to identify and localize regions of phenotypic interest (for example, to molecular oncologic pathologists), the alignment of the landscapes of different genes for comparing their spatial expression signatures, demarcate those areas that reveal gene expression above (or below) a certain level for focused molecular analysis (e.g., test for specific hallmarks of cancer), and characterize overall intratumor diversity. The interactive version of all the 3D plots is available at this project’s GitHub webpage (https://mortezahaji.github.io/Landscape-Project/ (accessed on 15 September 2022)).

GATHER analyzes spatial differential gene expression in single-cell transcriptomic data using two different approaches. An illustrative example is provided using a selected set of five CAF phenotypes, which were represented by the expression of the corresponding marker genes (the respective phenotypes are noted in parentheses): *CXCL12* (CAF-S1), *FBLN1* (mCAFs), *C3* (inflammatory CAFs), *S100A4* (sCAFs), and *COL11A1*, which is a fibroblast-specific “remarkable biomarker” that codes for collagen 11-α1 and shows expression gain in CAFs [88]. For details on the CAF markers, see reviews, e.g., [89,90].

In the first approach, at each point p of the tumor space, GATHER computes the differential expression of each of the above CAF genes between two sets of samples drawn from spatial neighborhoods that are (i) near to p versus (ii) distant from p using a semi-parametric 2-sample test for single cell data based on the 2-Wasserstein distance [55]. It outputs p-values obtained from the test, which are then adjusted for false discovery rate (FDR) by the Benjamini–Hochberg method. This allows for GATHER to map the locally significant CAF phenotypes. Figure 3 shows a 3D snapshot of the differentially expressed CAF genes at each point. For the lists of all the differentially expressed genes based on the above approach, see Table 1.

In the absence of ancillary external data, we performed an internal validation of the lists of differentially expressed genes as identified by GATHER based on a pathway over-representation analysis. Using this popular approach, the genes in Table 1 were input for unsupervised identification of statistically over-represented pathways from the well-known KEGG database. To conduct this test of over-representation, the DAVID platform was used. The results are shown in Appendix A.

The most recurrent (across the 10 lists) among the over-represented KEGG pathways (*p*-value < 0.001) that were identified by DAVID based on the results of GATHER are (a) protein digestion and absorption, (b) ECM–receptor interaction, and (c) focal adhesion. Clearly, each of these pathways are relevant to the known effects of fibroblasts on tumor microenvironment. Importantly, fibroblasts bio-chemically and bio-physically modify the extracellular matrix (ECM), such as by the production of ECM-degrading proteases (including matrix metalloproteinases, or MMPs) that degrade tumor-associated ECM, thereby remodeling the ECM to support tumor growth [91]. Furthermore, CAFs are known to create a nutrient-rich microenvironment via local stromal generation of mitochondrial fuels which are then absorbed and utilized by cancer cells to metabolically support tumor growth [92].

The two other pathways, i.e., the ECM–receptor interaction and the focal adhesion, are particularly noteworthy for their therapeutic potential in cancers. ECM modulates the hallmarks of cancer and multiple genes of this pathway are known to be expressed in breast cancer [93,94]. In particular, surface receptors interact with ECM components and ECM-bound factors to mediate cell adhesion and intracellular signaling [91]. This regulates multiple key processes in cancer cells such as proliferation, differentiation, migration, and apoptosis [95]. The prevention and elimination of cancer metastasis could play a significant role in the reduction of cancer-related deaths. In this regard, anti-cancer therapy can take advantage of the findings of the past decade that determined the significance of ECM remodeling at each stage of metastasis development [91].

In the context of cell motility and invasion, focal adhesion kinase (FAK) has been shown to be involved in the functional interplay among various mediators such as MMPS, mesenchymal markers, and focal adhesions [96,97]. Since its discovery in the early 1990s, FAK has been studied for its role in the regulation of cell spreading, adhesion, migration, survival, proliferation, differentiation, and angiogenesis [98]. Indeed, high FAK expression and activation are associated with poor clinical outcomes and metastatic features in breast cancer [98]. Thus, several preclinical studies have targeted FAK, a non-receptor tyrosine kinase, either using FAK inhibitors in combination with other chemotherapies or with immune cell activators [99].

In the second approach, GATHER partitions the tissue space into regions according to different levels of enrichment of a phenotype of interest, and identifies all genes that are expressed differentially across these regions. The regions of the landscape are characterized by l=3 discrete levels of each CAF phenotype: high (CAF z>1), mid (CAF 0.5<z≤1), and low (CAF z≤0.5). The levels provide graded spatial contexts in which certain genes may express differentially. Again, we used the 2-Wasserstein distance method to identify the differentially expressed genes across (a) the high CAF versus the medium CAF regions; and (b) the medium CAF versus the low CAF regions. Table 2 lists the genes that were thus found to be significantly differentially expressed across the spatial levels of CAF phenotypes.

Furthermore, we used GATHER to compute the spatial enrichment z scores {SEZ(S,p)|SϵC} for a collection C of hallmark gene sets of cancer as shown in Table 2. For the given tumor, the landscapes defined by the enrichment scores of each selected hallmark of cancer are described in 3D in Figure 4. In addition, the pointwise dominant hallmark, i.e., the gene set in C having the highest spatial enrichment z score at any given point, was determined and its distribution is depicted in 3D in Figure 5.

For comparative analysis, we employed two commonly used methods for RNAseq data analysis, namely, edgeR [83] and DESeq2 [84], to identify the differentially expressed genes in our input tissue sample. The top 10 most significant differentially expressed genes as obtained from these two aspatial methods are listed in the Appendix A. Appendix A maps the expression of some of these identified significant genes over the tissue space. Clearly, the lack of the capability of using spatial information by these two otherwise popular methods leads to very poor spatial signal in the output. This clearly demonstrates the advantage of a geostatistical modeling-based platform such as GATHER for the analysis of spatial single-cell gene expression data, especially in terms of its ability to detect neighborhoods with differentially expressed pathways (including hallmarks of cancer) specific to phenotypic contexts, and thus mapping the heterogeneity in the tumor microenvironment.

GATHER computes Batty’s spatial entropy index with a variable partitioning of the tissue space to output a quantitative measure of ITH. The tissue space is randomly partitioned into a fixed number of (G) polygons multiple (N=100) times. For each iteration, the spatial entropy is computed, which results in a bar plot for each choice of G. This is shown in Figure 6 for the spatial entropy of the expression of the *TBX3* gene in the given tumor sample. While the median spatial entropy tends to decrease as the heterogeneity is likely to reduce within smaller polygons generated by higher values of G, we select the first value of G for which the median entropy appears to flatten out as the optimal number of partitions. For the present example, the partition into $G^{*}=7$ polygons is selected, and thus, GATHER outputs Batty’s spatial entropy measure $H_{B}$(*TBX3*) as 0.942.

Importantly, the spatial heterogeneity of molecular signatures may be more insightful in the presence of a particular phenotypic context in a given tumor. To capture this with a quantitative measure, GATHER also computes the spatial co-occurrence based on Leibovici’s entropy measure. This allows the user to define phenotypic contexts within which selected genes or gene sets may express significantly. We illustrate this using six contexts as defined by five CAF phenotypes (as described above) and a sixth context (namely, “None”) wherein none of those phenotypes occur significantly. We test their co-occurrence with the enrichment of the selected hallmarks of cancer.

At each point p of the tissue space, the thresholds for the expression Z(C, p) of the dominant CAF phenotype C as well as SEZ(G, p) of the hallmark gene set *G* were set at 0.5. Taking combinations of the different CAF marker genes and cancer hallmark gene sets, the spatial heterogeneity of their co-occurrence is mapped. At each point, the combination with the most dominant phenotype is depicted. The map in Figure 7 uses the following colors to represent the (CAF, hallmark) pairs: red (*FBlN1*, PI3K_AKT_MTOR), blue (*C3*, Angiogenesis), purple (*COL11A1*, PI3K_AKT_MTOR), and grey (no significant CAF phenotype). The regional diversity of co-occurrence is clearly visible, which could be further analyzed by selecting other combinations with the platform’s interactive 3D visualization tool.

## 4. Discussion

In the 19th century, Rudolph Virchow, the “father of modern pathology”, observed the pleomorphism of cancer cells within tumors. In the 1970s, G.H. Heppner, I.J. Fidler, and others showed the existence of distinct subpopulations of cancer cells within tumors, which differed in terms of their tumorigenicity, their resistance to treatment, and their ability to metastasize. ITH has been shown to be associated with poor outcome and decreased response to cancer treatment in multiple human cancer types, implying a universal role in therapeutic resistance [100,101]. To this day, the quantitative assessment of cell-to-cell variation in the expression of a therapeutic target at the protein level is still a challenging task, which partly explains why explicit measures of ITH are not yet commonly used for guiding clinical decisions.

As noted above, GATHER has many practical advantages. The kriging estimates are based on a geostatistical model that allows GATHER to predict the expression value of a gene at any point of the transcriptomic landscape, which allows it to be represented as a surface that is both high-resolution and continuous. Thus, such landscapes can be visualized by an isopleth or contour map. Importantly, GATHER also computes and maps the standard errors of the gene-specific kriging estimates. Furthermore, as the gene-specific landscapes are synthesized over a common grid, they can be easily aligned and systematically combined to produce surfaces that can depict the spatial enrichment of gene sets or pathways of interest. Moreover, a quantitative measure of error associated with the kriging predictions is available as a spatial measure of quality—and mappable at every location—of the transcriptomic landscape. As yet another advantage, since the kriging prediction at any point is based on every available observation in any given neighborhood, the synthesis of a gene’s expression landscape by GATHER is in general not affected by the missing value problem that commonly afflicts single-cell RNAseq data.

Invasive and metastatic tumors often present with thorough tissue disorganization, leading to a microenvironment defined by cellular and paracrine interactions that allow for the selection and diversification of certain phenotypes that are not otherwise observed. For instance, blood and lymphatic vasculature in tumors are disorganized with significant functional, spatial, and temporal heterogeneity [102,103]. The resulting variability in nutrients, oxygenation, growth factors, and pH [104] can lead to various abnormal contextual signals that are absent in normal, healthy tissues. While spatial phenotypic contexts have been challenging to capture precisely with traditional approaches, high-resolution landscapes constructed by GATHER allow for the easy demarcation of such regions with the expression levels—above a selected cutoff—of the often well-characterized markers of these contexts.

In the present study, we used different CAF phenotypes as illustrative examples. The significance of such phenotypes could be understood from several experimental models of breast cancer and human tumors that reveal the spatial separation of the CAF subtypes attributable to different origins, including the peri-vascular niche, the mammary fat pad, and the transformed epithelium. Indeed, not only do the cancer cells and CAFs share location-specific signaling pathways, but the gene expression profiles for each CAF subtype indicate distinctive functional programs and hold independent prognostic capabilities in clinical cohorts by association to metastatic disease. GATHER is able to effectively identify at the single-cell level the genes with significant differential expression across the diverse spatial contexts as defined by the complex phenotypes that occur in heterogeneous tumor microenvironments.

Notably, an innovative quantitative feature of GATHER is its use of spatial entropy measures to evaluate ITH in a given tumor sample. It computes Batty’s entropy to evaluate the distribution of a particular phenotype, as determined by the expression of the corresponding markers, over a given tissue area. Furthermore, as a tissue area could provide the locations for more than one phenotype or the expressions of multiple markers of a complex phenotype, GATHER also computes a co-occurrence-based spatial entropy measure based on Leibovici’s. The randomization over the tissue space allows the resulting spatial entropy to yield a robust measure of ITH.

Next-generation genetically engineered mouse models can more accurately mimic human cancers [105], new multiplex immunostaining techniques, digital pathology, and specialized computational platforms are able to provide a more accurate quantitative assessment of ITH. New approaches such as MIBI [106] and cycIF [107] conduct assays on intact tissue samples, thereby maintaining tumor topology and cellular contexts. New computational approaches have also been developed to use next-generation sequencing data to assess ITH and infer the clonal evolution of a tumor. Although such techniques could be used to identify the different subpopulations of cells in a tumor’s microenvironment using a “parts-list” approach, it is much harder to clearly dissect the complex phenotypes of tumor cells in terms of their corresponding spatial contexts. Towards this, GATHER provides an efficient solution by substituting the approach of clustering discrete cells with each of their stochastically variable gene expressions by constructing continuous transcriptomic landscapes via a long-established geostatistical modeling approach.

We note that our present work has some limitations. The assumption of stationary mean by ordinary kriging may not always hold true in real data, although the method is known to yield relatively unbiased estimates despite non-stationarity [108]. Alternatively, other approaches such as universal kriging may be implemented in future work. GATHER does not explicitly group the cell subtypes as clusters like some of the other scRNAseq analysis tools do, although the expression landscapes of the known markers for different cell subtypes could be used to demarcate the regions that are enriched above a certain threshold and thus yield the cells therein. In our earlier papers, we developed a linear combination test (LCT) that can rigorously test for the enrichment of gene expression in a pathway against multivariate, continuous phenotypes of samples as opposed to univariate, binary outcomes used for traditional gene set analysis [109,110,111]. Recently, we extended the LCT to conduct a single-cell gene set expression analysis but without using spatial phenotypes [109]. In our future work, we will extend the LCT to test the enrichment of pathways across complex spatial phenotypes based on the capability of GATHER to analyze tissue heterogeneity in a given sample.

## 5. Conclusion 

Our computational platform, GATHER, is a new addition to the emerging field for spatial single cell omic data analysis. It introduces the use of geostatistical modeling for synthesis of high-resolution and continuous gene expression landscapes of a given tumor sample. Such landscapes allow GATHER to map the enriched regions of molecular pathways of interest in the tumor space and identify genes that have spatial differential expressions at locations representing specific phenotypic contexts using measures based on optimal transport. GATHER provides novel applications of spatial entropy measures that could be used for quantification and objective characterization of ITH.

## Figures and Tables

**Figure 1 cancers-14-05235-f001:**
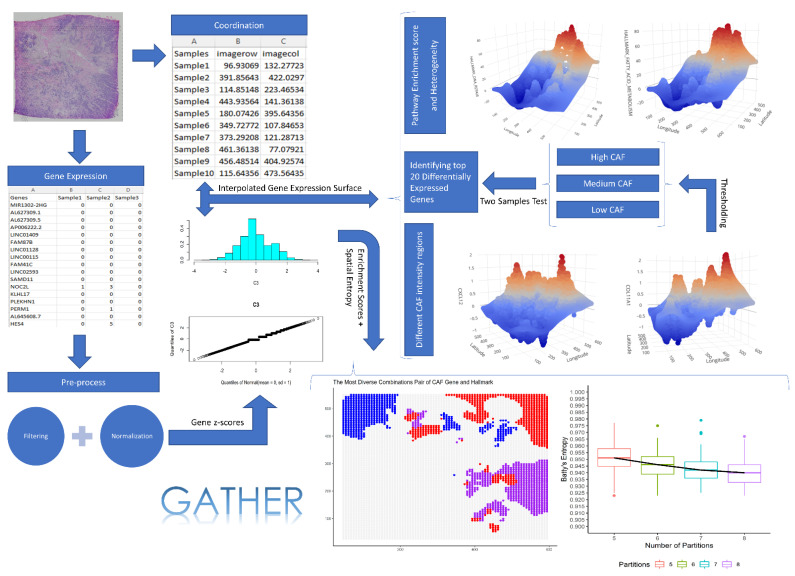
The GATHER workflow. It outlines the different analytical steps taken by GATHER starting from single-cell omics data preparation including normalization and filtering to the generation of kriging-predicted gene expression landscapes as well as iterative computation of spatial entropy measures. It also illustrates the interactive 3D visualization using GATHER of the computed gene- and gene set-specific landscapes defined over the input tissue space.

**Figure 2 cancers-14-05235-f002:**
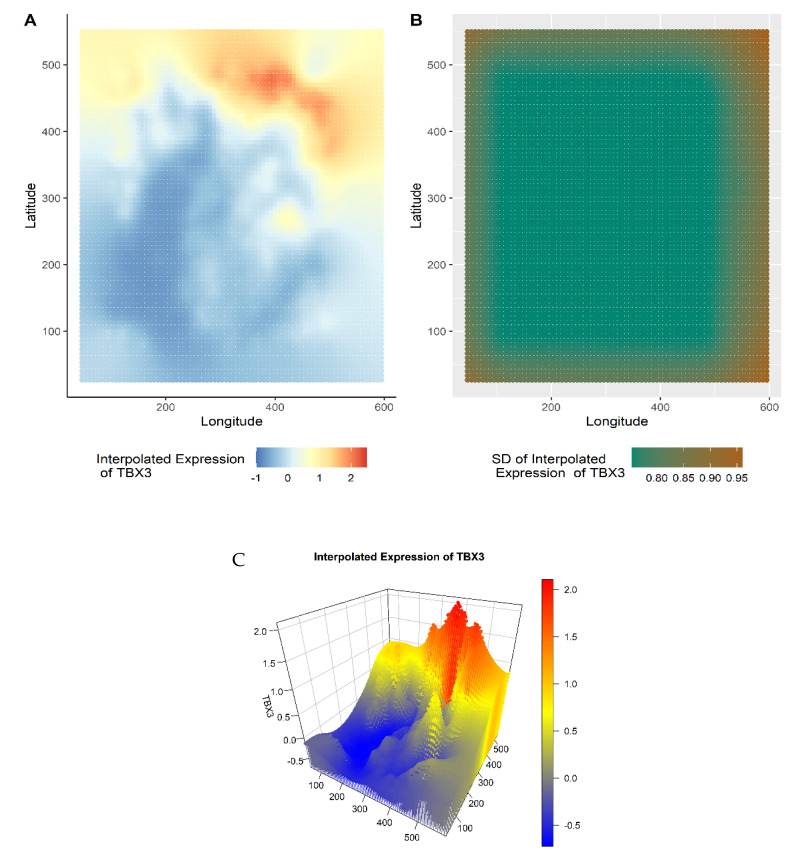
Gene expression landscape generated by geostatistical modeling. Taking the gene *TBX3* as an example, plots (**A**) and (**B**) show kriging-predicted value Z of gene expression at each point of the tissue space and the associated standard error, respectively. Plot (**C**) is a snapshot of the interactive 3D visualization of the plot (**A**). The x and y dimensions define the tissue space while the z dimension in plot (**C**) represents the kriging-predicted expression.

**Figure 3 cancers-14-05235-f003:**
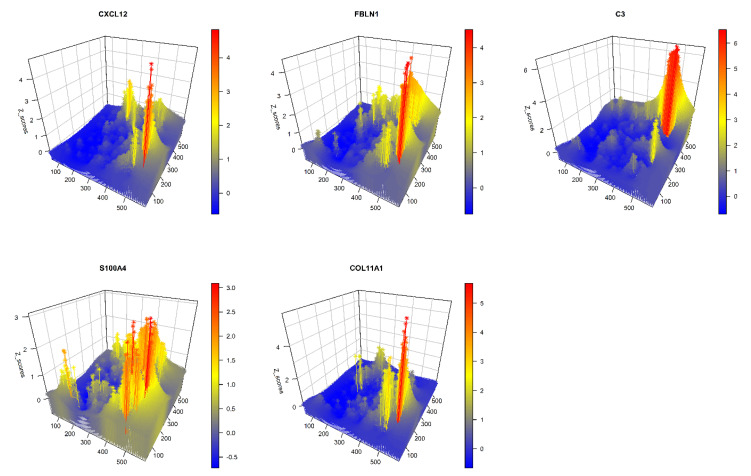
A 3D plot of gene-specific continuous transcriptomic landscapes of marker genes of different CAF phenotypes. The name of each CAF gene appears over its plot. The x and y dimensions define the tissue space while the z dimension represents the kriging predicted expression value (Z) at each point of the tissue space.

**Figure 4 cancers-14-05235-f004:**
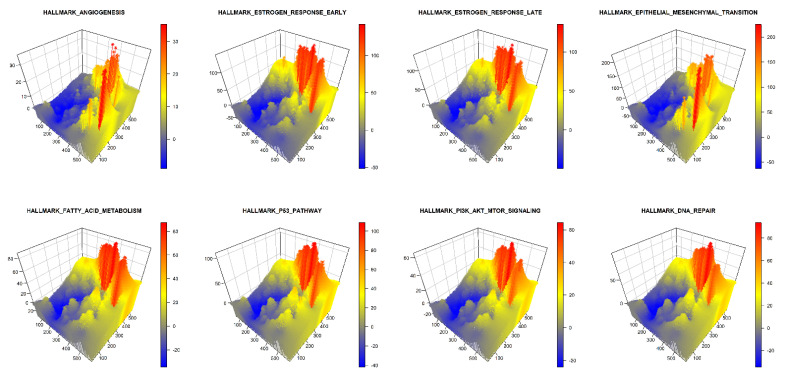
A 3D snapshot of the spatial enrichment z scores for different hallmark gene sets of cancer. The x and y dimensions define the tissue space while the z dimension represents the spatial enrichment z score (SEZ) at a given point. The name of each hallmark gene set appears over its plot.

**Figure 5 cancers-14-05235-f005:**
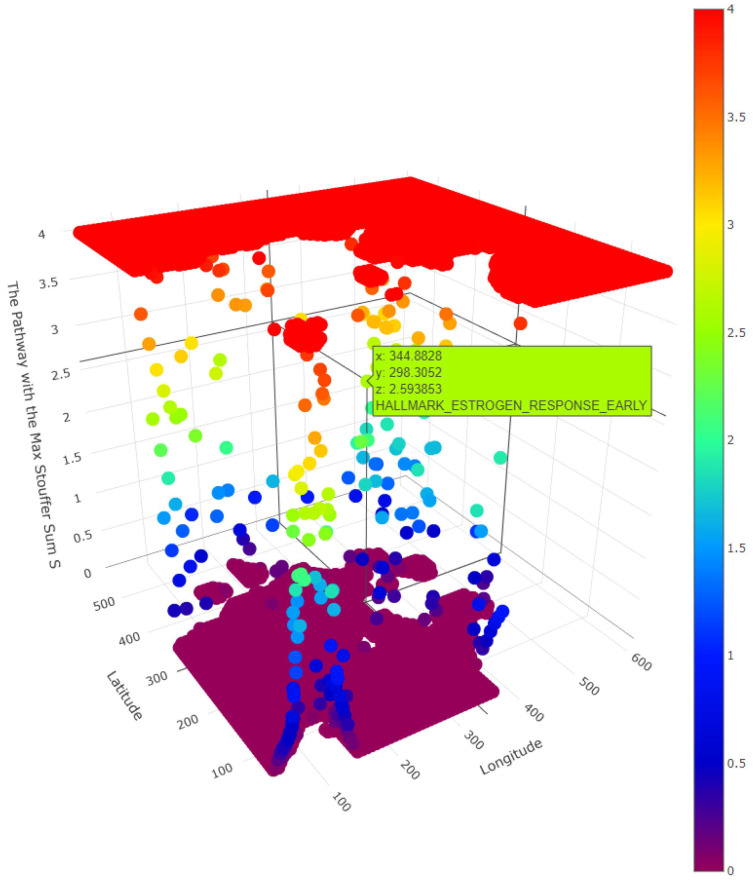
A 3D snapshot of the pointwise dominant hallmark gene sets of cancer. The x and y dimensions define the tissue space while the z dimension represents the maximum spatial enrichment z score (SEZ) at a given point among the selected hallmarks. One such point where PI3K_AKT_MTOR hallmark is dominant is shown as an example.

**Figure 6 cancers-14-05235-f006:**
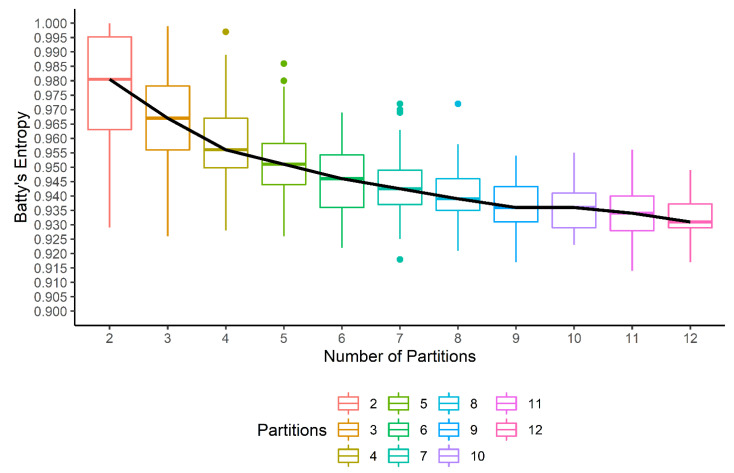
Batty’s spatial entropy as a measure of intratumor heterogeneity of gene (*TBX3*) expression. For different number of partitions (*x* axis) of the tissue space, N=100 spatial entropy values are calculated (*y* axis) and shown with a boxplot. The trend of the median entropy values is shown with a black line.

**Figure 7 cancers-14-05235-f007:**
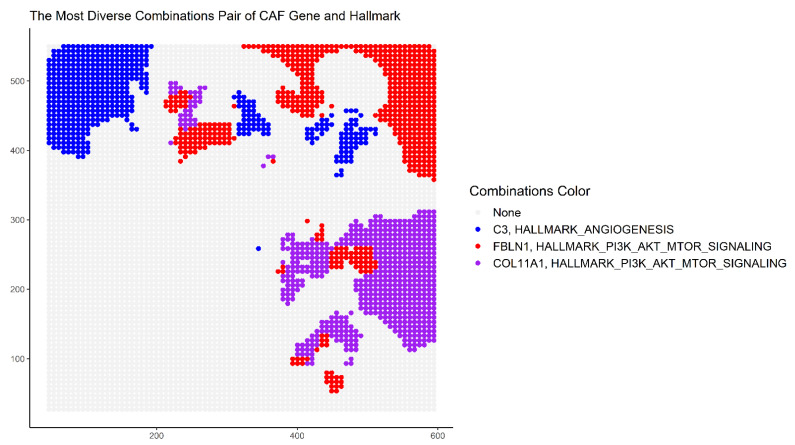
The co-occurrence based on Leibovici’s spatial entropy index. Taking combinations of the different CAF phenotypic contexts and cancer hallmark gene sets, the spatial heterogeneity of their co-occurrence is described. At each point, the combination with the most dominant phenotype is depicted. The colors used to represent the (CAF marker, hallmark gene set) pairs are red (FBlN1, PI3K_AKT_MTOR), blue (C3, Angiogenesis), purple (COL11A1, PI3K_AKT_MTOR), and grey (no significant CAF phenotype).

**Table 1 cancers-14-05235-t001:** Differentially expressed genes in five different CAF phenotypic contexts and their spatial entropy.

CAF Marker	High CAF Z ≥ 1 N	Medium CAF 0.5 < Z < 1 N	Low CAF Z ≤ 0.5 N	The Top 20 Most Common Expressed Genes in 100-Times Permutation at q < 0.2 (N = 50 Random Samples for All Groups)		Median of Batty’s Spatial Entropy
				High CAF vs. Medium CAF	Medium CAF vs. Low CAF	
COL11A1	190	3600	535	MMP11, COL1A2, FN1, DCN, S100A6, CTSK, COL3A1, COL1A1, TIMP3, LUM, SDC1, B2M, S100A4, COL10A1, LGALS1, COL5A2, SERPINF1, SPARC, HLA.A, CTSD	COL1A2, ASPN, DCN, SDC1, LGALS1, COL1A1, SPARC, TAGLN, HTRA3, POSTN, COL5A1, PRSS23, AEBP1, CALD1, ACTA2, COL5A2, PTMS, FN1, COL6A2, FSTL1	0.983
S100A4	223	3600	502	LGALS1, S100A6, COL3A1, ACTB, HTRA1, S100A10, TAGLN, COL6A3, CD74, CRABP2, POSTN, TMSB10, HLA.DRB1, PALLD, CLU, SPARC, COL1A1, PTMS, COL6A1, SDC1	FSTL1, SERPING1, COL3A1, COL6A2, FTL, ISLR, LGALS1, S100A6, SPARC, TAGLN, C1S, CILP, COL1A1, COL6A1, DCN, FLNA, HLA.DPA1, HLA.DPB1, PCOLCE, PTMS	0.982
CXCL12	141	3553	631	COL6A2, DCN, MMP2, HSPG2, NBL1, SERPING1, SERPINF1, COL6A1, ISLR, AEBP1, ASPN, SPARC, LUM, COL5A2, THY1, LRP1, COL1A1, MMP11, COL3A1, RARRES2	ACTB, ASPN, BGN, CALD1, CILP, COL1A1, COL3A1, COL5A1, COL6A2, DCN, FLNA, FN1, FSTL1, HTRA3, LGALS1, LUM, S100A6, SDC1, SPARC, TAGLN	0.983
C3	206	3501	618	HLA.DRA, FTL, CYBA, HLA.DPB1, APOE, HLA.DPA1, CD74, A2M, RPL13, IFI27, LAPTM5, TYROBP, CTSB, VIM, ACTB, HLA.E, SERPING1, HLA.DRB1, PSAP, TMSB10	APOE, COL5A1, FSTL1, SPARC, BGN, COL5A2, GPRC5A, PRCP, AP2M1, EDF1, HLA.DPA1, PITX1, ARHGAP1, COL6A1, COL6A2, CYB561, ATP5IF1, CD81, COL1A1, COL1A2	0.983
FBLN1	288	3449	588	LUM, COL3A1, COL6A2, FTL, C3, IFI27, COL1A1, COL1A2, MMP2, SERPING1, COL6A1, LRP1, SERPINF1, COL6A3, LGALS1, SPARC, FN1, ACTB, HTRA1, IFITM3	COL3A1, DCN, SPARC, CILP, COL5A1, FN1, LGALS1, MYL9, ACTB, ASPN, CALD1, COL1A1, COL6A2, MMP11, POSTN, S100A6, TAGLN, TPM4, COL1A2, COL6A1	0.982

**Table 2 cancers-14-05235-t002:** The hallmark gene sets of cancer selected for this study.

Gene Sets		Number of Genes in Gene set	Overlap with the Gene List of the Study (%)	Overlap among the 8 Hallmark Gene Sets							
				1	2	3	4	5	6	7	8
1	HALLMARK_EPITHELIAL_MESENCHYMAL_TRANSITION	201	81 (40%)	-	2%	<1%	<1%	0	0	<1%	<1%
2	HALLMARK_ANGIOGENESIS	37	12 (32%)	2%	-	0	0	0	0	0	0
3	HALLMARK_ESTROGEN_RESPONSE_EARLY	201	64 (32%)	<1%	0	-	8%	<1%	0	<1%	<1%
4	HALLMARK_ESTROGEN_RESPONSE_LATE	201	62 (31%)	0	0	8%	-	0	0	<1%	1%
5	HALLMARK_DNA_REPAIR	151	42 (28%)	0	0	<1%	0	-	0	0	<1%
6	HALLMARK_PI3K_AKT_MTOR_SIGNALING	106	28 (26%)	0	0	0	0	0	-	0	<1%
7	HALLMARK_FATTY_ACID_METABOLISM	159	41 (26%)	<1%	0	<1%	<1%	0	0	-	<1%
8	HALLMARK_P53_PATHWAY	201	50 (25%)	<1%	0	<1%	1%	<1%	<1%	<1%	-

## Data Availability

Data are contained within the article.

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
