# Peer review of "Geostatistical Modeling and Heterogeneity Analysis of Tumor Molecular Landscape"

_cancers, 2022, doi:10.3390/cancers14215235_

Round 1
Reviewer 1 Report
This well written paper is an interesting summary of the evolution of concepts regarding the representation of the intratumor heterogeneity in relation with the emergence of innovative tools of analysis. In this article the authors focus on the concepts derived from the spatial omic information via a new computational platform named after its acronym GATHER.
Questions and remarks are as follows:
- «GATHER is able to effectively identify at single cell level the genes with significant differential expression across the diverse spatial contexts as defined by the complex phenotypes that occur in heterogeneous tumor micro-environments ». What is the definition of « complex phenotypes » in the context of this work? Is it based on histology, immunohistochemistry? The authors should show the integration of gene expression, spatial location AND histolopathology/immunohistochemistry.
- Validation of the approach: this article is based on anonymized data available for secondary analysis. The authors should validate their method using one or several clinical specimens and show the resulting annotations of cancer cells and CAFs
- How does GATHER compare with other advanced algorithms to process the datasets?
- Technically, what kind of samples and how these samples can be used for extracting datasets to be linked to GATHER?
- Can the GATHER workflow be implemented on molecular biology platforms in oncology departments. In other terms how does the GATHER workflow complement the pathological annotation of tumors?
- « From GATHER to bedside »: The authors should pay attention to the translational perspectives in terms of tumor druggability, personalized treatment, or prognosis. In particular they should provide a graphical abstract focused on a workflow guide illustrating the oncological relevance of GATHER in « real life ».
Reviewer 2 Report
The present article, "Geostatistical Modeling and Heterogeneity Analysis of Tumor Molecular Landscape" written by Pyne et.al., is a well explained study about a computational platform GATHER. This study is relevant in the field of ITH with respect to gene expression landscapes of tumor samples. It would be interesting to observe the proficiency/ potential of GATHER in single cell gene expression analysis across complex spatial phenotypes.
Author Response
We thank Reviewer 2 for reviewing our manuscript. We have noted in the manuscript the results, including the lists of differentially expressed genes, due to GATHER across different complex spatial CAF phenotypes.
Round 2
Reviewer 1 Report
I see no reason to delay further the publication of this article.